# Antimicrobial and Immunomodulatory Potential of Cow Colostrum Extracellular Vesicles (ColosEVs) in an Intestinal In Vitro Model

**DOI:** 10.3390/biomedicines10123264

**Published:** 2022-12-15

**Authors:** Samanta Mecocci, Livia De Paolis, Roberto Zoccola, Floriana Fruscione, Chiara Grazia De Ciucis, Elisabetta Chiaradia, Valentina Moccia, Alessia Tognoloni, Luisa Pascucci, Simona Zoppi, Valentina Zappulli, Giovanni Chillemi, Maria Goria, Katia Cappelli, Elisabetta Razzuoli

**Affiliations:** 1Department of Veterinary Medicine, University of Perugia, 06123 Perugia, Italy; 2National Reference Center of Veterinary and Comparative Oncology (CEROVEC), Istituto Zooprofilattico Sperimentale del Piemonte, Liguria e Valle d’Aosta, Piazza Borgo Pila 39-24, 16129 Genova, Italy; 3S.C. Biotecnologie Applicate alle Produzioni, Istituto Zooprofilattico Sperimentale del Piemonte, Liguria e Valle d’Aosta, via Bologna 148, 10154 Torino, Italy; 4Department of Comparative Biomedicine and Food Science, University of Padova, 35020 Legnaro, Italy; 5S.C. Diagnostica Generale, Istituto Zooprofilattico Sperimentale del Piemonte, Liguria e Valle d’Aosta, via Bologna 148, 10154 Torino, Italy; 6Department for Innovation in Biological, Agro-Food and Forest Systems (DIBAF), University of Tuscia, 01100 Viterbo, Italy

**Keywords:** colostrum, Extracellular Vesicles, colibacillosis, coloEVs, antimicrobial

## Abstract

Extracellular Vesicles (EVs) are nano-sized double-lipid-membrane-bound structures, acting mainly as signalling mediators between distant cells and, in particular, modulating the immune response and inflammation of targeted cells. Milk and colostrum contain high amounts of EVs that could be exploited as alternative natural systems in antimicrobial fighting. The aim of this study is to evaluate cow colostrum-derived EVs (colosEVs) for their antimicrobial, anti-inflammatory and immunomodulating effects in vitro to assess their suitability as natural antimicrobial agents as a strategy to cope with the drug resistance problem. ColosEVs were evaluated on a model of neonatal calf diarrhoea caused by *Escherichia coli* infection, a livestock disease where antibiotic therapy often has poor results. Colostrum from Piedmontese cows was collected within 24 h of calving and colosEVs were immediately isolated. IPEC-J2 cell line was pre-treated with colosEVs for 48 h and then infected with EPEC/NTEC field strains for 2 h. Bacterial adherence and IPEC-J2 gene expression analysis (RT-qPCR) of *CXCL8*, *DEFB1*, *DEFB4A*, *TLR4*, *TLR5*, *NFKB1*, *MYD88*, *CGAS*, *RIGI* and *STING* were evaluated. The colosEVs pre-treatment significantly reduced the ability of EPEC/NTEC strains to adhere to cell surfaces (*p* = 0.006), suggesting a role of ColosEVs in modulating host–pathogen interactions. Moreover, our results showed a significant decrease in *TLR5* (*p* < 0.05), *CGAS* (*p* < 0.05) and *STING* (*p* < 0.01) gene expression in cells that were pre-treated with ColosEVs and then infected, thus highlighting a potential antimicrobial activity of ColosEVs. This is the first preliminarily study investigating ColosEV immunomodulatory and anti-inflammatory effects on an in vitro model of neonatal calf diarrhoea, showing its potential as a therapeutic and prophylactic tool.

## 1. Introduction

Colostrum is the first unique nutrition source for the mammalian newborn, giving nutritive elements and playing an important role in health protection and the development of the immune system. Cow’s colostrum, compared with milk, is enriched in fat, proteins, peptides, non-protein nitrogen, ash, vitamins and minerals, hormones, growth factors, cytokines and nucleotides, which influence metabolism, endocrine systems and the nutritional state of neonatal calves. Moreover, it contains enzymes with antimicrobial functions and immunoglobulins conferring passive immunity, and it stimulates the development and function of the gastrointestinal tract [1]. The sophisticated signalling system of maternal messages to promote postnatal health is also composed of transcription modulatory elements, such as small RNAs, derived from various cell sources of the mammary gland and present in different milk fractions (cells, lipids, and the skim milk) [2]. Small RNAs can be found as free molecules or packed in vesicles that confer protection to these labile molecules, allowing to overcome the harsh conditions of the gastrointestinal tract [3]. 

Extracellular Vesicles (EVs) are micro- and nano-sized phospholipidic double-layer-enclosed systems, mainly functioning as message delivery vectors of molecules, such as proteins, antigens, lipids, metabolites, RNAs and, as recently found, DNA fragments [4,5,6]. Being produced by all cell types, EVs can be taken-up by receiving cells, inducing the regulation of biological processes though the molecular cargo delivery [7]. EVs can be classified as: exosomes, released from the fusion of multi-vesicular bodies and cell membrane, generated through the late endosomal pathway and characterized by the smallest size (30–150 nm); microvesicles (ectosomes or shedding vesicles), released for exocytosis and ranging from 100 to 1000 nm; and the biggest apoptotic bodies (generally greater than 1000 nm) [5,7,8,9,10,11]. However, the impossibility to distinguish the different types merely on the basis of their size induced the International Society of Extracellular Vesicles (ISEV) to adopt and suggest the use of a size-based nomenclature, namely small, medium/large EVs and apoptotic bodies [12]. Being released in the extracellular environment, EVs can be recovered from every biological fluid, including milk (mEVs) and colostrum (colosEVs) [13,14,15,16,17].

The amount of EVs and dimension-based subtypes within colostrum/milk depend on the lactation period, characterized by a dilution in the EVs concentration going through the lactation stages and an enrichment in small EVs in colostrum [18]. The lactation stage also influences the cargo composition and function, with colosEVs particularly characterized by proteins and transcripts that could be related to the regulation of homeostasis and the immune system at a protein–protein interaction functional analysis, although further studies are needed to prove this hypothesis [19]. ColosEVs have been found to induce apoptosis in cultured cells [20] and liver cancer cells and to decrease the gene expression of inflammatory and angiogenetic genes [21]. Similar to bovine mEVs, colosEVs showed a positive effect on inflammation and immune regulation in colitis [22] and were able to shift the inflammatory process to the fibroblast proliferation, migration and endothelial tube formation, accelerating wound recovery [23]. A role in shaping microbiota was found for cow colosEVs, which correlates with bone remodelling and partial recovery in a osteoporosis mouse model [24]. Bovine transcriptomic analysis shows differences between the colosEVs of two breeds, thus highlighting variability among individual features and the need for deeper investigations into these important natural signalling systems [25]. 

Neonatal calf diarrhoea represents a relevant livestock disease caused by different types of *Escherichia coli* (*E. coli*), for which antibiotic therapy often has poor results due to the high rates of multi- and extensively drug-resistant isolates [26,27]. To date, it represents one of the major health challenges for livestock production, since it can lead to an increased death rate and, subsequently, to serious economic losses [28]. In this context, colosEV features can be exploited for alternative strategies in fighting infections, assisting or replacing the action of conventional antimicrobials [29] when trying to cope with the multi-drug resistance problem [30]. Antimicrobial properties have been found for EVs deriving from different human cells but also from plant and animal products, such as honey [31,32,33]. This study aimed to investigate the protective potential of colosEVs using an in vitro model of neonatal calf diarrhoea. In detail, we evaluated the immunomodulatory, anti-inflammatory and antimicrobial effects of ColosEV pre-treatment on porcine intestinal epithelial cells (IPEC-J2) infected with wildtype *E. coli* isolates. According to previous studies, IPEC-J2 was proven to be an effective in vitro model for investigating host–pathogen interactions [34] since it is able to express and produce cytokines, toll-like receptors (TLRs) and mucins [35]. By infecting ColosEV-pre-treated IPEC-J2 cells, we intended to highlight the effect of ColosEV on bacterial adhesion and immune-related gene expression and, thus, to preliminarily investigate the feasibility of ColosEV application as prophylaxis against infectious disease in cattle herds.

## 2. Materials and Methods

### 2.1. Farm and Animals Selection

In order to minimize variables capable of influencing the EV composition (stress, nutrition, genetics, age), selection criteria allowing to have a homogeneous study group of cows were adopted. Five heifers (primiparous cows) were selected from a closed-cycle farm consisting solely of Piedmontese cattle with the presence of newborn calves affected by neonatal diarrhoea attributable to ETEC. Selected animals were fed in the pre-calving period with a homogeneous diet.

### 2.2. Strains Isolation and Characterization

Following episodes of diarrhoea in new-born calves, stool samples and deceased subjects were transferred to the laboratories for strain isolation. A total of 1 g of faeces was diluted in 10 mL of buffered peptone water BPW (Biolife, Monza, Italy) and then submitted to serial 10-fold dilution in the same medium until reaching the fifth dilution. The last three dilutions were streaked by a 10 μL loop onto McConkey agar plates (MCK) (Microbiol, Macchiareddu, Italy) and incubated overnight at 37 °C. Thirty suspected *E. coli* colonies (relatively big, rounded, convex and pink colonies on purple-turned medium) were selected from the three MCK plates, sub-cultured on blood agar plate (BA) and MCK, then incubated overnight at 37 °C and 42 °C, respectively. Identification procedures based on growth on MCK at 42 °C, indole production and Gram staining suggestive of *E. coli* were performed before the preparation of six pooled samples with selected colonies. The pooled samples were inactivated by heat and submitted to molecular test aimed to highlight presence of some pathogenic factors specific for: enteropathogenic *E. coli* (EPEC) intimin-coding gene (*eae*); ETEC adhesin (K99) and shigatoxin genes (*STa*, *STb*); and/or necrotoxigenic *E. coli* (NTEC) cytotoxic necrotizing factor 1 (*CNF1*). Colonies selected for biomolecular analyses were diluted in 200 µL of ultrapure water and subjected to DNA extraction by thermal lysis (boiling for 15 min on heating mantle). The extracted DNA was subjected to qualitative–quantitative analysis by BioSpectrometer^®^ (Eppendorf, De). The determination of *eae* presence was performed by a multiplex PCR according to ISO/TS 13136 [36], while the identification of *K99*, *STa* and *STb* was carried out with a multiplex PCR described by Casey et al. (2009) [37]. Moreover, *CNF1* was detected in simplex PCR. Primers, amplicon sizes and thermal profiles are summarized in Appendix A. When the pooled samples gave positive results for target pathogenic factors, the single colonies constituting the positive pool [38] were submitted to the cited molecular test in order to confirm the results and obtain the strain eligible for drug susceptibility test. Drug susceptibility was tested using the minimum inhibitory concentrations (MICs) method according to the guidelines of the Clinical and Laboratory Standards Institute (CLSI). Quality controls of the plates used for MIC were performed according to the CLSI VET01S supplement (CLSI, 2020) [39]. MIC breakpoints (expressed in μg/mL) were evaluated and interpretative criteria were retrieved from both human (M100) [39] and veterinary CLSI Standards (VET08) [40] from the European Committee on Antimicrobial Susceptibility Testing (EUCAST) [41] and from the guidelines of the Committee for the Antibiotic Susceptibility testing of the Société Française de Microbiologie Comité de l’Antibiogramme (CA-SFM) [42] (Appendix A).

Collected *E.coli* strains were used for further in vitro assays (see Section 2.5.2 and Section 2.6). 

### 2.3. Colostrum Collection and ColosEV Isolation

From each of the five heifers, three samples of colostrum collected within 24 h of calving were pooled to reach a final quantity of 500 mL and stored at 4 °C for less than 24 h before processing, avoiding cryo-preservation to minimize artifacts. ColosEVs were isolated by serial differential centrifugations (DC) and a step using ethylenediaminetetraacetic acid tetrasodium salt dihydrate (EDTA) before an ultracentrifugation at 35,000× *g* for 1 h at 4 °C. The resulted supernatant was submitted to ultracentrifugation at 200,000× *g* for 90 min at 4 °C using a Beckman Coulter Optima L-100 XP with a 45 Ti rotor following the protocol of Mecocci et al. [43]. Collected colosEV pellets were used for morphological analysis and in vitro assays. 

### 2.4. ColosEV Morphological Characterization

#### 2.4.1. Western Blotting

To extract proteins, colosEVs were lysed in RIPA Buffer (25 mM Tris-HCl pH 7.6, 150 mM NaCl, 1% NP-40, 1% sodium deoxycholate, 0.1% SDS). Total protein concentration was measured using Bradford assay. A quantity of 25 µg of total proteins were separated using 12% T sodium dodecyl sulfate–polyacrylamide gel electrophoresis and transferred to polyvinylidene fluoride membranes. Blotted membranes were saturated and incubated overnight with primary antibodies against CD81 (1:500; Bioss Antibodies, Woburn, MA, USA), TSG-101 (1:400, Santa Cruz Biotechnology, Santa Cruz, CA, USA) and calnexin (1:400 sc-23954, Santa Cruz Biotechnology, Santa Cruz, CA, USA). Anti-rabbit HRP-conjugated IgG (1:3000, Cell Signaling Technology, Danvers, MA, USA) and anti-mouse HRP-conjugated IgG (1:3000, Cell Signaling Technology) were used as secondary antibody. Clarity Western ECL Substrate (Bio-Rad) was used to evidence protein bands, and the images were acquired using a GS-800 imaging systems scanner (Bio-Rad, Hercules, CA, USA).

#### 2.4.2. Transmission Electron Microscopy (TEM) 

Few drops of colosEV suspension were deposited on Parafilm. Formvar-coated copper grids (Electron Microscopy Sciences) were gently placed on the drops with the coated side towards the suspension and colosEVs were allowed to adhere to the grids for about 20 min. Grids were contrasted with 2% uranyl acetate for 5 min after being rinsed in PBS and distilled water. The observation was performed using a Philips EM208 transmission electron microscope equipped with a digital camera (University Centre of Electron and Fluorescence Microscopy—CUMEF).

#### 2.4.3. Nanoparticle Tracking Analysis (NTA)

A Malvern Panalytical NanoSight NS300 nanoparticle tracking analysis (NTA) system (Malvern, Worcestershire, UK) was used to assess the concentration and size distribution of isolated colosEVs. One colosEV pellet, derived approximately from 15 mL of colostrum, was resuspended and diluted in filtered (0.22 μm pore size) phosphate-buffered saline (PBS) (Sigma, St. Louis, MO, USA) to be suitable for the NTA system’s working concentration range, and five measurements were performed. Concentration and diameter results are reported as mean ± 1 standard error of the mean.

### 2.5. Cell Cultures

IPEC-J2 cells (porcine jejunal epithelial cells, IZSLER Cell Bank code BS CL 205) were grown in a mixture (1:1) of Dulbecco’s Modified Eagle (DMEM) (Euroclone, Milan, Italy) and Nutrient Mixture F-12 (F12) (Euroclone, Milan, Italy) enriched with 10% Fetal Bovine Serum (FBS, GIBCO™, Thermofisher scientific, Milan, Italy), 1% L-glutamine solution (Carlo Erba Reagents s.r.l., Milan, Italy) and 1% penicillin/streptomycin solution (Carlo Erba Reagents s.r.l., Milan, Italy). We decided to use these cells because they spontaneously secrete IL-8 and were previously employed in studies on bacterial and virus pathogenicity and on the intestinal inflammatory response [44]. 

#### 2.5.1. Cell Viability Assay

Firstly, in order to determinate the most suitable vesicle concentration to be used for the in vitro assay, we tested different colosEV quantities at scalar concentrations in terms of protein weight (0.015 μg, 0.15 μg, 1.5 μg, 15 μg, 150 μg). A 2,3-bis-(2-methoxy-4-nitro-5-sulfophenyl)-2H-tetrazolium-5-carboxanilide (XTT) assay was performed according to the manufacturer’s instructions (XTT Cell Viability Kit, Cell Signaling Technology Inc., Danvers, MA, USA). In brief, IPEC-J2 cells were plated on a 96-well plate (100 µL per well, 0.1 × 10^5^) in DMEM/F12 medium, 10% FBS, 1% L-glutamine and 1% penicillin/streptomycin and incubated for 24 h at 37 °C until confluence. The day after, the 96-well plates’ seeding cells were exposed to a scalar concentration of colosEVs and then incubated again for 48 h at 37 °C, 5% CO_2_. Untreated cells were used as controls. XTT assay was performed at selected time points (24 h and 48 h). On the day of the assay, cell culture medium was removed and replaced with 100 µL of fresh DMEM/F12 medium supplemented with XTT detection solution (1:50). The plates were then incubated again at 37 °C, 5% CO_2_ for 2 h, and the absorbance was measured at 450 nm using a multimode microplate reader (Glomax^®^, Promega™, Milan, Italy). This assay was performed three times for each colostrum at scalar concentrations, setting up four technical replicas each.

#### 2.5.2. ColosEV and *E. coli* Cell Treatments

Cells were seeded into 12-well culture plates (1 mL per well, 1.5 × 10^5^ cells/mL), as stated in Section 2.5, and exposed to 1.5 µg (protein weight) colosEVs, incubated at 37 °C, 5% CO_2._ After 48 h, 1 mL of *E. coli* suspension was added to the cultures and incubated at 37 °C in 5% CO_2_ for 2 h. For *E. coli* stimulation, a pool of two different *E. coli* strains (EPEC, NTEC) was used. Briefly, bacteria were grown in brain heart infusion (BHI) medium at 37 °C for 12–18 h. Subsequently, 200 μL of overnight cultures were inoculated into fresh BHI and incubated for 1–2 h at 37 °C in order to obtain mid-log-phase cultures. *E. coli* strains were pelleted and resuspended as 10^8^ colony-forming units (CFU)/mL in DMEM/F12, 1% L-glutamine solution. 

At 2 h post-stimulation bacteria were removed and cell monolayers were gently washed five times with DMEM/F12, 1% L-glutamine solution. Untreated cells were used as control. The experiment was performed three times with three technical replicates each. 

For each experiment, we have untreated cells (“Control” in the graphs), cells infected with *E. coli* strains (“*E. coli*” in the graphs) and cells pre-treated with 1.5 µg (protein weight) colosEVs and subsequently infected with *E. coli* (“*E. coli* + colosEVs” in the graphs).

### 2.6. Bacterial Adhesion Assay

For the experimental point reported in Section 2.5.2, cells were lysed by adding 200 μL of 1% Triton X-100 in PBS at room temperature for 5 min [44,45,46]. After cell lysis, 800 μL of PBS was added to each well; the resulting cell suspension was vortexed, serially diluted and seeded on Tryptone Bile X-GLUC (TBX) agar plates. 

### 2.7. Gene Expression Assay 

Cells of the experimental points reported in Section 2.5.2 were also tested for gene expression. After removing supernatants, cells were washed three times and again incubated in their medium for 3 h. Cells were lysed with 400 μL of RLT buffer (Qiagen, Hilden, Germany) and, after incubation for 10 min at room temperature, they were collected and stored at −80 °C until analysis. Untreated cells were used as controls.

Total RNA was extracted from IPEC-J2 cells using RNeasy Mini Kit (Qiagen s.r.l., Milan, Italy) following manufacturer’s instructions and eluted in 50 µL of ultrapure RNase-free water. RNA extracted was quantified through Qubit 3.0 Fluorometer (Thermo Fisher Scientific, Waltham, MA, USA).

From each sample, 250 ng of RNA was reverse-transcribed into cDNA, using iScript^®^ cDNASyntesis Kit (Bio-Rad, Milan, Italy). Real-time PCR amplification was performed on CFX96™ Real-Time System (Bio-Rad, Milan, Italy) in accordance to a protocol previously described [47,48].

Primers of target and reference genes were derived from previous studies [49,50] or designed according to the sequences available on the Primer-BLAST online design platform (https://www.ncbi.nlm.nih.gov/tools/primerblast/, accessed on 31 July 2021). Primer pairs were placed in different exons or at exon–exon junctions to avoid biases due to genomic DNA amplification. Specific primer pairs for the reference genome were verified in silico using in silico PCR software (https://genome.ucsc.edu/cgi-bin/hgPcr, accessed on 31 December 2021) to confirm their specificity for targeting. Primer sequences of target genes and gene reference are reported in Table 1. 

### 2.8. Statystical Analysis

A Kolmogorov–Smirnov test was carried out to check Gaussian distributions in the data sets concerning viability assay, gene expression and bacterial adhesion.

All the results failed the Kolmogorov–Smirnov test; therefore, statistically significant differences were checked through the non-parametric Kruskal–Wallis test, followed by a Dunn’s post hoc test for viability assay and gene expression. For the bacterial adhesion assay, CFU data were converted into log10 values and a Student’s *t*-test was performed. The significance threshold was set at *p* < 0.05 (Prism 5, GraphPad Software, GraphPad Software Inc., San Diego, CA, USA).

## 3. Results

### 3.1. Strains Isolation and Characterization

Two different *E. coli* strains were isolated from stool samples. The strain n.1, which resulted positive for the presence of the intimin-coding gene (*eae*), was identified as EPEC; differently, the strain n.2 was classified as a chimeric NTEC since it resulted positive for *eae* and the *CFN1* gene (Appendix A). Results obtained from antimicrobial characterization through MIC are summarized in Appendix A. Isolated *E. coli* strains were used for further in vitro assays.

### 3.2. ColosEVs Morphologic Characterization

The presence and purity of colosEVs in the pellet generated from the colostrum along with the size range and shape assessment were determined, respectively, by Western blotting and TEM. At TEM, colosEVs appeared rather homogeneous in shape. They ranged from 30 to 120 nm and were single or arranged in aggregates. The absence of cell debris and background confirmed the efficiency of purification procedure (Figure 1A,B). Western blot assay proved the EVs presence, showing a positive reaction for Tumor Susceptibility Gene 101 protein (TSG101) and Cluster of Differentiation 81 (CD81). One vesicle-negative antigen, the calnexin, was used to evaluate the presence/absence of cellular contaminants (Figure 1C). Based on the NTA data, the colosEV mean (±standard error) diameter was 177.4 ± 2.4 nm (Figure 1D). The distribution of colosEV populations showed three main picks at different diameters (110 nm, 172 nm and 253 nm) and a mode of 117.5 ± 5.3 nm (D10 = 105.0 ± 0.9 nm and D90 = 269.7 ± 4.8 nm). NTA also determines nanoparticle densities, reporting the mean concentration (particles/mL) (±standard deviation) of five measurements after a pellet resuspension in 400 μL of PBS that resulted in 1.02 × 10^12^ (±4.88 × 10^10^).

### 3.3. Cell Viability 

The IPEC-J2 cells cultured in 96-well plates and treated with scalar concentrations of colosEVs (see Section 2.5.1) were analysed by measuring the absorbance at 450 nm using a microplate reader. As shown in Figure 2, 150 µg (protein weight) colosEVs at 48 h determined a significant reduction in cell viability with respect to controls (*p* < 0.0001). No significant effect in terms of cell viability was observed at 24 h (Appendix A).

### 3.4. Gene Expression Assay 

In order to determine the effect of colosEVs on cells infected with *E. coli* strains, we compared IPECJ2 gene expression in *E. coli*-inflamed cells, pre-treated cells with 1.5 µg colosEVs and then infected with *E. coli* and untreated ones (control). According to the results obtained, a significant upregulation of *CXCL8* (*p* < 0.001), *NFKB1* (*p* < 0.001), *DEFB1* (*p* < 0.001), *DEFB4A* (*p* < 0.001), *RIGI* (*p* < 0.01) and down-regulation of *IFNB* (*p* < 0.05) was highlighted in *E. coli*-inflamed cells compared with control (Figure 3). 

Concerning cells pre-treated with colosEVs and then infected, our results highlighted a significant decrease in *TLR5* (*p* < 0.05), *CGAS* (*p* < 0.05) and *STING* (*p* < 0.01) gene expression compared with *E. coli*-inflamed cells, while no significant modulation was proved for the other tested genes (Figure 3). 

### 3.5. Bacterial Adhesion

Since NTEC/EPEC strains are associated to neonatal calf diarrhoea in cattle [27], we performed our in vitro bacterial adhesion assay by using previously isolated *E. coli* strains (see Section 3.1). In detail, our aim was to investigate the ColosEV effect on host–pathogen interactions and, more specifically, to evaluate if ColosEV pre-treatment could modulate *E. coli* in vitro adhesion. Based on our results, the pools of strains under study were able to adhere to IPECJ2 cells. Our results highlighted greater adhesion of *E. coli* strains on untreated IPEC-J2 (7.5 ± 0.06 SD log10 CFU/5 × 10^5^) compared with cells pre-treated with 1.5 µg colosEVs cells (7.3 ± 0.09 SD log10 CFU/5 × 10^5^) (*p* = 0.006) (Figure 4). In particular, exposure to 1.5 µg colosEVs cells determined a decrease in bacterial adhesion of 0.2 log10 CFU/10^5^ cells.

## 4. Discussion

In order to reduce the use of bactericidal and bacteriostatic chemotherapeutic agents in animals, which is necessary to also fight the spread of antibiotic resistance in humans, an increasing attention to research aimed at clarifying the mechanisms of action underlying both innate and acquired immunity should be considered strategic, since there is a need to obtain new tools capable of enhancing current prophylactic and therapeutic protocols. 

Calf neonatal diarrhoea infections, caused by several strains of *E. coli,* are among the livestock diseases where the use of chemotherapeutic agents is the most common treatment practice [27,51]. However, antimicrobial-based therapeutic protocols achieve poor results due to the widespread of antibiotic resistance; the immunization of the calves takes place through the intake of colostrum and failure of transfer of passive immunity to the calf entails a high risk of mortality. 

This study provides insights into the interaction between pathogens–hosts and the possibility to provide potential therapeutic and prophylactic support using colosEVs that are known to contain a miRNA cargo capable of modifying the responses of the calf’s innate immunity, as described by Hata and collaborators [52]. In this study, the presence of RNA capable of modulating the early differentiation of lymphocytes and regulating inflammatory processes was found for milk and, even more so, for colostrum EVs. 

Our results evidenced how the cellular adhesion of *E. coli* induces modifications of inflammation-related gene expression. Indeed *CXCL8* and *NFKB1* were significantly up-regulated after bacterial incubation, while *IFNB* was down-regulated [53]. These data are in agreement with previous studies on *E. coli* infection. In particular, Tsai and co-workers demonstrated that *E. coli* adhesion determines the increase in *CXCL8* expression by the activation of MAPK and NF-kB pathways [54]. Our results highlighted the increased expression of the RIGI gene, which is one of the molecules responsible for the transcription factor NFKB1 pathway activation in *E. coli* infections [55]. *E. coli* infection also induces the inhibition of IFN-β production, promoting inflammation and barrier disruption [56], a condition that seems to be also established in our experiment. 

Moreover, the *E. coli* adhesion of our model determines the up-regulation of *DEFB1* and *DEFB4A* transcripts whose encoded proteins may function as important regulators of host defence against exogenous pathogens. Indeed, these cationic antimicrobial peptides have broad-spectrum antibacterial activity, inhibiting the growth of *E. coli* and modulating the innate immune response [57]. 

Conversely, an antimicrobial reaction through the down-regulation of *TLR5, CGAS* and *STING* genes was seen in *E. coli-*infected cells pre-treated with colosEVs. It is worth noting that *TLR5* is the host molecule that recognizes *E. coli* flagellin and activates the immune response [58]. Indeed, the *E. coli* infection modifies *TLR5* distribution and increases its presence on the cell surface through EPEC flagellum, translocation of effectors and intimate adherence; in turn, the modulation of *TLR5* leads the intimate adherence that alters the proinflammatory response [59]. In our study, *TLR5* was up-regulated after *E. coli* infection as a consequence of both *NFKB1* and *CXCL8* increases; furthermore, 48 h of colosEV incubation significantly down-regulated *TLR5*, demonstrating the capabilities of these vesicles to interact with the molecular mechanism behind the *E. coli* intimate adhesion.

Beyond *TLR5* activation, innate cytosolic sensing of foreign nucleic acids represents an important trigger of innate immune responses in several microbial infections. There are mechanisms of crosstalk between the cytosolic RIG-1 and cGAS–STING nucleic acid-sensing pathways that amplify the innate antimicrobial responses against both RNA and DNA [60]. Indeed, a central regulator of cytosolic DNA sensing is cyclic GMP-AMP (cGAMP) synthase (cGAS), and cGAMP binds to an essential cytosolic sensor stimulator of interferon genes (STINGs). STINGs, moreover, antagonize RIG-1 by binding its N-terminus, probably to avoid the overactivation of RIG-1 signalling and the associated autoimmunity [61]. Nevertheless, most studies have focused on viruses’ infections, and only a few studies evaluated the interaction between bacteria, especially *E. coli*. To date, the cGAS–STING signalling pathway has emerged as a key mediator of inflammation in the settings of infection, cellular stress and tissue damage, and has recently emerged as a nodal player in immunity that is currently being explored as a potential therapeutic target [62]. 

In this contest, in our experimental study, *CGAS* and *STING* genes were up-regulated after *E. coli* incubation, as expected, and, interestingly, down-regulated in cultures pre-treated with colosEVs for 48 h and then infected. This suggests the colosEVs and the molecular cargo may be inhibitors of the cGAS-STING pathway and therefore responsible for the decreased bacterial adhesion observed in our experiments after vesicle treatment (Figure 4). Nevertheless, epithelial cells participate actively in the innate immune response in the gut. Thus, we may speculate that their stimulation, for example by the EV cargo, might result in the generation of antibacterial bioactive molecules to eliminate E. *coli*. Our findings are particularly relevant considering that recent studies have linked STING and type I IFNs to necroptosis, a highly pro-inflammatory form of cell death [63,64]. The down-regulation of this pathway by colosEVs suggests their anti-inflammatory activity.

Concerning host–pathogen interaction, as mentioned above, colosEV pre-treatment determines the reduction in *E. coli* adhesion, probably due to the decrease in *TLR5* expression and the down-regulation of the cGAS/STING pathway. In this way, we can speculate that adhesion reduction could be due to the modulation of cellular receptor expression. It is worth noting that the strains isolated in this study showed multi-antimicrobial resistance; therefore, the evidence that colosEV treatment altered the host–pathogen interaction suggests the possibility of using them as an alternative to antibiotic treatment.

## 5. Conclusions

The results obtained from our study highlighted the capacity of colosEVs to modulate host–pathogen interactions. In particular, our data showed a significant reduction in the ability of *E. coli* to adhere to cells pre-treated with 1.5 µg colosEVs when compared with untreated IPEC-J2. Furthermore, a significant decrease in *TLR5*, *CGAS* and *STING* gene expression was detected, suggesting that ColosEV molecular cargo could exert an antimicrobial activity by modulating the cGAS/STING pathway, a key player in inflammation, tissue damage and cellular response. This is a preliminary but promising result that highlights a possible mechanism of action of the molecular cargo against microbial infections. It represents a starting point for future studies to further evaluate the feasibility of colosEV application as a natural antimicrobial agent to cope with the drug resistance problem.

## Figures and Tables

**Figure 1 biomedicines-10-03264-f001:**
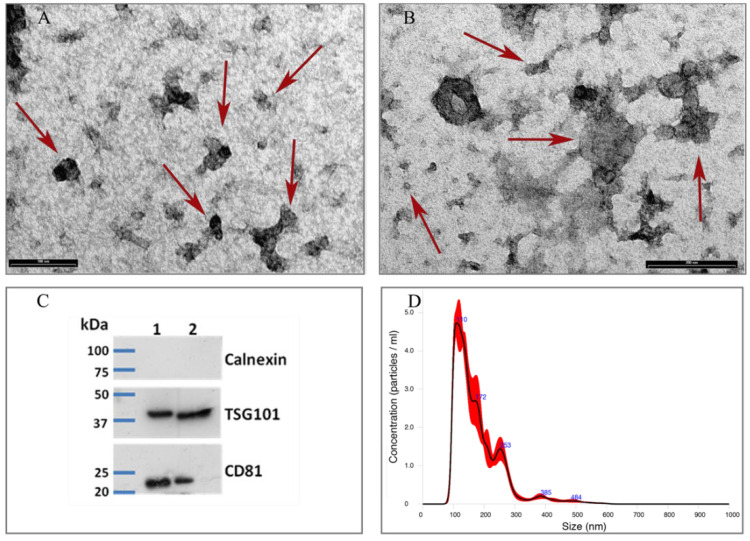
Morphological characterization of isolated colosEVs: Transmission electron microscopy low (**A**) and high (**B**) magnification micrographs showing single and clustered colosEVs indicated by red arrows. Scale bar: A. 100 nm; B. 200 nm; (**C**) Western blot images obtained using Ab against Tsg101 (Tumor Susceptibility Gene 101 protein) and CD81 (Cluster of Differentiation 81) that are both mEV antigens and calnexin as negative cellular debris control of two colosEV samples (1 and 2); (**D**) nanoparticle tracking analysis graph indicating colosEV size distribution.

**Figure 2 biomedicines-10-03264-f002:**
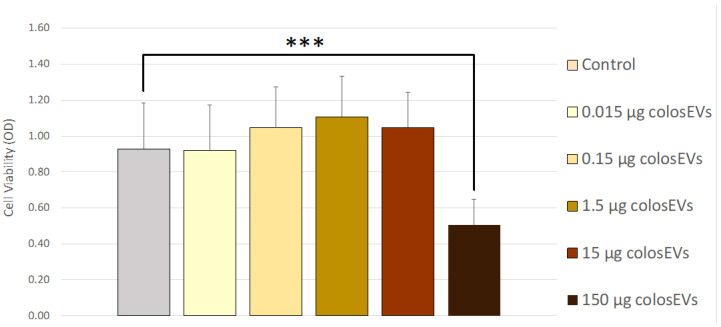
Viability of IPEC-J2 after colosEVs exposure at 48 h. The different concentrations of colosEVs did not determine a significant difference in terms of cell viability after 24 h (see Appendix A), whereas 150 µg colosEVs determined a significant reduction in IPECJ2 vitality at 48 h (*p* < 0.0001, indicated by *** in the graph). Data are expressed as optical density (OD) ± SD. Differences were evaluated through the Kruskal–Wallis test and applying the post hoc Dunn’s multiple comparison test.

**Figure 3 biomedicines-10-03264-f003:**
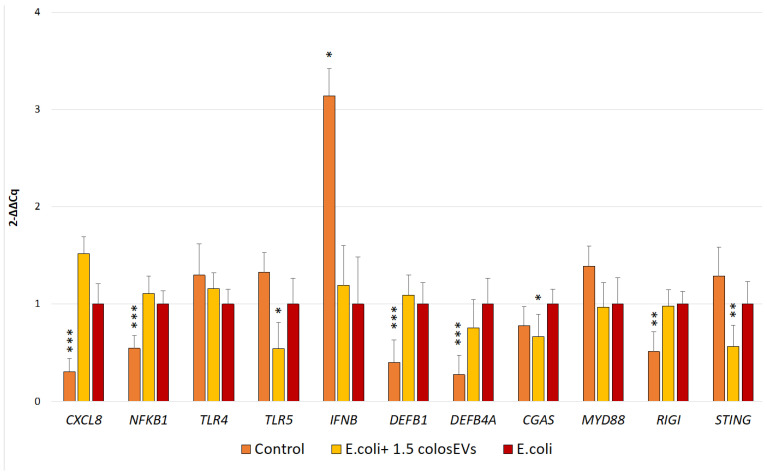
Effect of 48 h +1.5 µg colosEVs on *E. Coli*-inflamed IPEC-J2 cells. IPEC-J2-tested conditions were: inflamed (*E. coli*, pink), inflamed 1.5 µg colosEVs (*E. coli* + 1.5 µg colosEVs, light yellow), untreated (control, dark yellow). Significant differences are reported with respect to infected cells with *E. coli*. Differences were evaluated through the Kruskal–Wallis test and applying the post hoc Dunn’s multiple comparison test. The asterisks indicate the statistical significance: * *p* < 0.05, ** *p* < 0.01 and *** *p* < 0.001.

**Figure 4 biomedicines-10-03264-f004:**
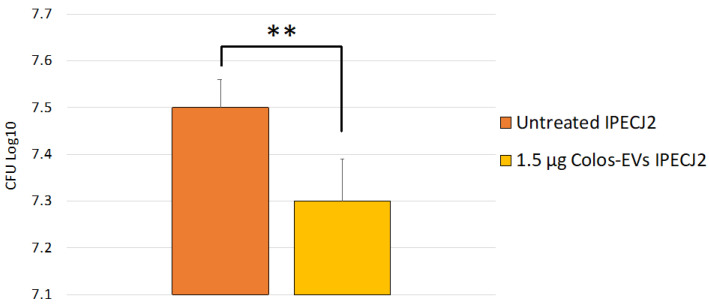
Adhesion of *E. coli* strains on 1.5 µg colosEVs pre-treated cells compared with untreated ones. Data are expressed as log10 CFU of adherent, *E. coli*/5 × 10^5^ cells and mean value of 3 experiments ± 1 standard deviation. Differences were evaluated through the Student’s *t*-test. The asterisks indicate the statistical significance: ** *p* < 0.01.

**Table 1 biomedicines-10-03264-t001:** Primer set sequences for target genes and reference.

Gene	Primer Sequences	Amplicon Length	Source
*CXCL8*	For—5′-TTCGATGCCAGTGCATAAATA -3′	175	[44]
	Rev—5′-CTGTACAACCTTCTGCACCCA-3′		
*NFKB1*	For—5′-CGAGAGGAGCACGGATACCA-3′	61	[44]
	Rev—5′-GCCCCGTGTAGCCATTGA-3′		
*TLR4*	For—5′-TGGCAGTTTCTGAGGAGTCATG-3′	71	[44]
	Rev—5′-CCGCAGCAGGGACTTCTC-3′		
*TLR5*	For—5′-TCAAAGATCCTGACCATCACA– 3′	59	[44]
	Rev—5′-CCAGCTGTATCAGGGAGCTT-3′		
*IFNB*	For-5′-AGTTGCCTGGGACTCCTCAA-3′	59	[44]
	Rev-5′-CCTCAGGGACCTCGAAGTTCAT-3′		
*DEFB1*	For—5′-CTGTTAGCTGCTTAAGGAATAAAGGC-3′	80	[44]
	Rev—5′-TGCCACAGGTGCCGATCT-3′		
*DEFB4A*	For—5′-CCAGAGGTCCGACCACTA-3′	87	[44]
	Rev—5′-GGTCCCTTCAATCCTGTT-3′		
*CGAS*	For—5′-TGGAGTGAAATGTTGCAGGAAAGA-3′	149	XM_013985148
	Rev—5′-GGGTCCTGGGTACAGACGTG-3′		
*STING*	For—5′-GCCTGCATCCATCCATCCCA-3′	226	MK302493.1
	Rev—5′-GCTGCTCTGGTACCTGGAGTG-3′		
*RIGI*	For—5′-GAATCTGCACGCTTTCGGGG-3′	96	NM_213804.2
	Rev—5′-CTGCACCTCATCGTCCCTA-3′		
*GAPDH*	For—5′-ATGGTGAAGGTCGGAGTGAA-3′	61	NM_001206359.1
	Rev—5′-AGTGGAGGTCAATGAAGGGG -3′		

## Data Availability

Not applicable.

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
