# Peer review of "Antimicrobial and Immunomodulatory Potential of Cow Colostrum Extracellular Vesicles (ColosEVs) in an Intestinal In Vitro Model"

_biomedicines, 2022, doi:10.3390/biomedicines10123264_

Round 1

Reviewer 1 Report

Although the quality of study is high some language corrections are needed and moreover I have few remarks on the content.

Line 130 - it is not clear where is Table S1 or is it part of Tbele 1 (which part)

Same for Table S2 - line 141

Line 32 - in vitro should be in italic

Line 48 cow should be cow's

Line 56 contained should be present

Line 57 packaged should be packed

line 69 basing should be based

Line 97 separate words whichantibiotics

Line 97 to high rates should be to the high rates

Line 110 gr should be g

Line 115 the three should be three

Line 119 the selected should be selected

Line 120 the presence should be presence

Line 130 Table S1?

Line 152 in vitro should be italic

Line 158 ug / correct unit

Line 185 insert they after word because

Line 209 weigh should be weight

Line 221 same as 209

Line 266 post should be post hoc

Line 276 Table S3?

Line 277 Table S4?

Line 281 exclude word of

Line 282 size should be the size

Line 309 post-doc should be post-hoc

Line 317 E. coli should be in italic

Line 320 seen should be proved

Lines 331 , 332 105 should be 105

Line 341 post-doc should be post-hoc

Line 344 In order to reduce....

Line 347 are strategic should be considered strategic.

Line 347 insert: There is a need to obtain ne tools ...

Line 352 widespread should be the widespread

Line 353 to transfer should be of transfer

Line 374 the growth should be growth

Line 382 resulted should be was

Line 395 evaluating the should be evaluated

Line 400 experimental results should be study

Line 412 the reduction should be reduction

Line 412 TLR5 should be the TLR5

Line 417 use this should be use them

Reviewer 2 Report

The manuscript entitled Antimicrobial and immunomodulatory potential of cow colostrum Extracellular Vesicles (colosEVs) in an intestinal in-vitro model” presents research work. But manuscript needs amendmnets on basis of following points to make this paper eligible to be published. I recommend the minor revision.

1)     The design of this study is not explained in effective manner, please explain the rational of this work at end of introduction

2)     Quantitative results are not presented in abstract and conclusion section. Please also explain the conclusion at end of abstract section

3)     Generally, the manuscript needs extensive English improvements.

4)     Please explain the statistical tools used in this study in experimental section?

Reviewer 3 Report

line 25: Z should appear before *, as in the order of list of authors

line 27-28: too long sentence,

Author tried to define exosomes and also describe the importance of exosomes.

as it is in abstract, short and precise sentences, may be carrying one message is advised.

lIne 76-78: the author https://doi.org/10.1016/j.jprot.2021.104338 described they determined proteins involved in porcine colostrum "may be" related to immunity and should be studied.

But not concluded the exosomes are involved. 

the author should correct the narrative of this sentence. 

Line 91: I think, the authors can use other scientific litratures, than using news as reference. 

If it is an avoidable, the person or expert described that scientific hypothesis should be mentioned. 

Line 204-205: Test performed in quadriplate?

5 colos and control is 6?

this should be properly written.

Line 221: what was the logic to treat the cells after treatment with colostrol.

As a therapy approach, treating infected cells would have a logical flow,

is there scientific reason or any drug test approach reference for this??

personally, it is not convenient.

line 226: cells lysed in triton X, for 6 minute?

It seemed longer time in my experience.

If the author did like that and if it is correct, it should be supported with references.

line 307: treatment with Colos EVS should enhance viability, not reduce. is that right?

Very importantly, the results indicates that the effect are not time dependent and not dose dependent. 

as doses of EVs increase from 0.015 to 150 micro grams, 1,000,000x, what is the logic?

when doses are changing, the effects are not accordingly; 0.015 reduce viability higher than 15 micro gram? 

Line313: The dose having significant difference from control, in viability is 150 micro gram, why author selected 1.5 micro gram?

this seems author use, dose which showed difference (in this case, the difference could be a bias, not effect of colosEXo

Line 423: This sentence is out of scope of the results of study, mechanism not addressed 

Round 2

Reviewer 3 Report

Well, the authors understand comments and addressed, the manuscript sound.